# The role of education in the association between self-rated health and levels of C-reactive protein: a cross-sectional study in rural areas of China

Raoping Tu,[1,2,3] Kuan-Yu Pan,[4] Guoxi Cai,[2,5] Taro Yamamoto,[2] Hui-Xin Wang[4,6]

For numbered affiliations see end of article.

**Correspondence to**
Dr Hui-Xin Wang;
huixin.wang@su.se

Dr Taro Yamamoto;
y-taro@nagasaki-u.ac.jp

## ABSTRACT

**Objectives** This study aims to examine the association between self-rated health (SRH) and levels of C-reactive protein (CRP) among adults aged 45 to 101 years old in rural areas of China, and to explore the role of education in the association.

**Design** Cross-sectional study.

**Setting** The study population was derived from two databases in China: Nanping project (NP) and the China Health and Retirement Longitudinal Study (CHARLS).

**Participants** There were 646 participants from a rural area of Nanping (NP) and 8555 rural participants from a national representative sample of China (CHARLS).

**Methods** CRP was measured using a high sensitivity sandwich enzyme immunoassay in the NP and immunoturbidimetric assay in the CHARLS. SRH was assessed by SRH questionnaires and categorised into good and poor. Education was measured by the maximum years of schooling and dichotomised into illiterate and literate. Multivariate linear regression models were used to study the associations.

**Results** Compared to people with good SRH, those with poor SRH had higher levels of CRP in NP (β=0.16, 95% CI −0.02 to 0.34) and in CHARLS (β=0.07, 95% CI 0.02 to 0.11) after adjusting for potential confounders. Similar findings were observed in the pooled population (β=0.08, 95% CI 0.03 to 0.12), especially in men (β=0.13, 95% CI 0.06 to 0.20) and in literate people (β=0.12, 95% CI 0.06 to 0.18).

**Conclusion** Poor SRH may be a predicator of elevated levels of CRP among middle-aged and older people in rural areas, especially in men and literate people.

## Strengths and limitations of this study

► Our study population came from two databases, including one national representative sample derived from the China Health and Retirement Longitudinal Study (CHARLS), making our results highly generalisable to the national rural population of China.

► C-reactive protein (CRP) was an objective measure performed by health professionals using validated methods, making it more reliable than subjective measures.

► Cross-sectional study design prevented us from making causal inferences.

► Convenience sampling in the Nanping project and the relatively large proportion of CHARLS participants with missing values in CRP may have introduced bias.

► Residual confounding or hidden bias cannot be ruled out due to lack of information on some potential confounders, such as clinical cardiovascular risk factors (e.g., High-density lipoprotein cholesterol(HDL-C); Hemoglobin A1c (HbA1c)), acute inflammatory conditions and medication use.

## INTRODUCTION

C-reactive protein (CRP), a marker of systemic inflammation, has been shown to be involved in crucial pathogenesis in a variety of negative health outcomes, including cardiovascular diseases,[1 2] diabetes,[3] cancer[4] and cognitive decline.[5] Since the value of CRP in the prediction of prognoses in health outcomes has been recognised, it is important, from a public health perspective, to identify people at risk of elevated CRP in an efficient and simple way.

Self-rated health (SRH) refers to an individual's subjective perception of his/her own health and can be easily measured. Despite this, SRH has been featured as a strong predictor for functional ability,[6] chronic diseases[7] and mortality.[8 9] Therefore, many health authorities have introduced SRH for surveillance.[10] The association between SRH and CRP has been examined in previous studies, but the results were inconsistent.[11–14] These discrepancies may be due to differences in characteristics of the study populations (e.g., age and sex) and study design. For example, a Japanese study demonstrated an association between poor SRH and an elevated CRP value in women, but not in men (age range 40–69).[14] In contrast, in an US sample of younger adults (mean age 28.42±1.78), current SRH was not associated with CRP in women, whereas the association was shown in

men.[13] Among hospital-based studies, poor SRH was associated with higher CRP in female patients with coronary heart disease,[12] but not in patients with breast cancer.[15] In community-based studies, there has been a cross-sectional association between SRH and CRP,[13 14] but no evidence indicating longitudinal association.[16]

As SRH measures personal perception of health, it can be influenced by other factors beyond the real health status. For example, people with different educational levels may have different perceptions of health.[17] This education-related difference in perception of health may further play a role in the association between SRH and health outcomes. Indeed, a stronger association between SRH and mortality among higher educated than lower educated individuals has been shown in two studies.[18 19] Since CRP has been recognised as an important predicator of mortality,[20] education seems to modify its relationship with SRH.[21] It is noteworthy that studies concerning the association between SRH and CRP were mostly conducted in developed countries where the study populations were relatively well educated.[11–14] To our knowledge, no study has focused on the difference in the association between SRH and CRP between illiterate and literate people. In China, despite the decrease in illiteracy from 1990 to 2010, there continues to be large difference between urban and rural areas: the rate of illiteracy in rural areas is two times more than that of urban areas.[22] Considering the lack of resources in rural areas, identifying people at risk of negative health outcomes using a simple measure such as SRH is warranted.

In this study, we use two databases from China to examine the association between SRH and CRP among middle-aged and older people in rural areas, and to explore whether the SRH-CRP association varies across age (45–60/≥60), sex (men/women) and educational levels (illiterate/literate).

## METHODS
### Study population
#### Nanping project (NP)
NP is a 2015, voluntary participation, cross-sectional study consisting of residents aged 18 years or older from one county of Nanping City in Fujian Province, China. Seven villages were selected based on recommendations from local health workers, since the residents in these areas are known to be highly cooperative.

As showed in figure 1, a total of 797 people were enrolled in the NP. To match with the age range of study population from the CHARLS, we excluded 98 participants under 45 years old. Those with CRP concentrations higher than 6.25 mg/L in dried blood spots (DBS), which is comparable to 10 mg/L at serum level[23] (n=25), were excluded due to potential acute inflammatory conditions. After further excluding people with missing information on CRP (n=2), SRH (n=25) and on both CRP and SRH (n=1), 646 people remained in this study.

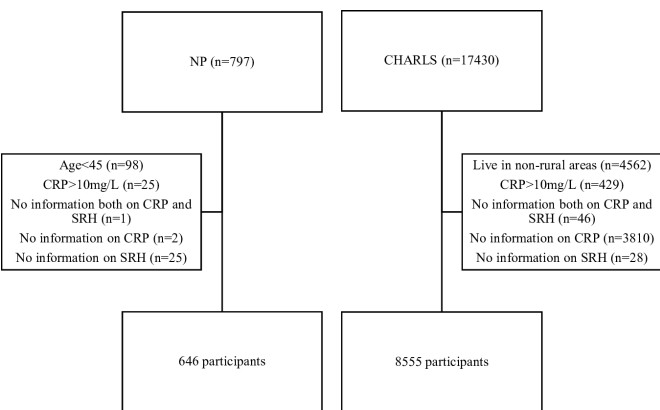

**Figure 1** Flow chart of the study populations in Nanping project and China Health and Retirement Longitudinal Study.

## China Health and Retirement Longitudinal Study (CHARLS)
The CHARLS is a nationally representative longitudinal study. Eligible people were selected through a multi-stage probability sampling, and detailed descriptions of sampling method are provided in the users' guide.[24] In this study, we used data from the baseline survey in 2011 because the CRP data were only available in that year. This is a secondary analysis of the CHARLS public database.

Overall, 17 430 people were examined at baseline (figure 1). People who lived in communities, or in both villages and communities (n=4562), and had CRP >10 mg/L (n=429) were excluded. We further excluded people with missing data on CRP (n=3810), SRH (n=28) and on both CRP and SRH (n=46). Finally, 8555 (69%) people were included in the analytical sample.

## Self-rated health (SRH)
SRH was assessed by one question: 'In general how would you rate your health?' Response options were 'good', 'average', 'poor' and 'very poor'.

## C-reactive protein (CRP)
### NP
Finger prick blood samples were collected by health workers using a filter paper, known as DBS. We kept the DBS at room temperature for a few days after being desiccated during the investigation period, then stored them in the Fujian Medical University at −20°. We used high sensitivity sandwich enzyme immunoassay method to measure CRP concentrations by applying monoclonal antibodies.[23] Further details of the protocols have been presented elsewhere.[25]

### CHARLS
The venous blood samples were collected by trained staff from local Chinese Center for Disease Control and Prevention (China CDC). Plasma samples were collected and preserved in 0.5 mL cryovial at −20°C, delivered to Beijing CDC within 2 weeks. Plasma CRP was determined by the immunoturbidimetric assay method at Capital Medical University.[26]

## Covariates

In both cohorts, all participants were interviewed face-to-face by trained interviewers using a questionnaire that covers information on age, sex, education, marital status, smoking, alcohol consumption and health status. Height and weight were measured by interviewers using standard anthropometers.

Education level was determined by maximum years of schooling: 0 year (illiterate), 1–6 years (elementary school), 7–9 years (junior high school), 10–12 years (senior high school) and >12 years (college or above). Due to the fact that more than 30% of both the NP and CHARLS samples were illiterate, we dichotomised education into 0 year (illiterate) and >0 year (literate). Age was dichotomised as 45–60 years versus ≥60 years old, and marital status as married versus non-married. Body mass index (BMI) was calculated by dividing weight (kg) by height squared ($m^2$) and categorised as underweight (<18.5), normal weight (18.5–24.99), overweight (25–29.99) and obese (≥30). Smoking was dichotomised into current smokers and non-current smokers (including former smokers). Alcohol consumption was categorised as regular drinkers (more than three times per week) and non-regular drinkers.

Health status was measured by asking the participants whether they had any moderate/severe disease symptoms (e.g., fever) in the last month, or used antihypertensive or antidiabetic medications in the NP, and whether they had ever been diagnosed by a doctor with any diseases (e.g., hypertension), or often suffered from any pain currently in CHARLS. People answering positively were categorised as unhealthy, otherwise healthy.

## Statistical analysis

First, data from the NP and CHARLS were analysed separately. We applied one-way analysis of variance (ANOVA) to examine the differences of CRP in characteristics in each data set by using F-distribution. The CRP variable was log-transformed because it was not normally distributed. The association between SRH and CRP was estimated by β-coefficient and a 95% CI using linear regression in two data sets. The first estimate was respective; in the second, data sets were pooled. Fixed-effect meta-analysis was used to examine the heterogeneity. Then we re-ran the linear regression using the pooled data set.

Age, sex and education were introduced into the basic-adjusted model. Further, we additionally adjusted for marital status, smoking, alcohol consumption, BMI and health status.[27 28] All analyses were repeated in the stratified analyses by age, sex and levels of education.

In order to compare our results with previous studies that included participant with formal education only, we performed additional linear regression analysis stratified by age and sex among illiterate and literate participants separately.

All statistical analyses were performed with Stata V.13.0 (Stata Corp).

## Patient and public involvement

There were no participants involved in the development of this study.

## RESULTS

### Characteristics of the participants

The CRP levels across different characteristics of participants were compared in each data set separately. Table 1 shows that in both data sets that people with advanced age, higher BMI, poorer SRH or an unhealthy status were more likely to have elevated levels of CRP. The findings were inconsistent with sex, education, marital status, smoking and alcohol consumption in the two data sets. People with missing CRP values in NP and CHARLS were better educated and reported better health status compared with those who remained in the analyses (online supplementary tables S1 and S2).

### SRH and CRP

Table 2 presents the association between SRH and CRP in the two individual populations. In the NP, a borderline statistically significant association was observed between very poor SRH and elevated levels of CRP (β=0.39, 95% CI −0.07 to 0.85) in basic-adjusted model, while the association was attenuated after adjusting for confounders (β=0.29, 95% CI −0.15 to 0.73). Despite insignificance, the estimated effect of SRH started to change direction from average SRH (β=−0.05) to poor SRH (β=0.10). In CHARLS, poor and very poor SRH were both associated with higher CRP (β=0.06, 95% CI 0 to 0.12; β=0.11, 95% CI 0.01 to 0.22). Considering the same pattern in both two data sets that poor and very poor SRH have similar effect on CRP and so as good and average SRH, and that there are limited number of participants with very poor SRH in NP, we combined 'good' and 'average' as good SRH, 'poor' and 'very poor' as poor SRH. Further, we found that poor SRH was associated with higher levels of CRP both in NP (β=0.16, 95% CI −0.02 to 0.34) and CHARLS (β=0.07, 95% CI 0.02 to 0.11) (table 2).

As the same direction of effect of estimate and a very low level of heterogeneity (I-squared <0.001%) were observed in the two data sets (data not shown), we pooled the data and re-ran the linear regression analyses in the combined populations. The association between poorer SRH and higher CRP was observed in the pooled population (β=0.08, 95% CI 0.03 to 0.12) (table 2).

### The roles of age, sex and education in the association between SRH and CRP

The association between SRH and CRP stratified by age, sex, education is shown in figure 2. In middle-aged people, worse SRH was associated with higher CRP both in NP (β=0.42, 95% CI 0.14 to 0.71) and CHARLS (β=0.06, 95% CI −0.01 to 0.12). Among older people, a similar trend was observed in CHARLS (β=0.08, 95% CI 0.02 to 0.15), but not in the NP. When stratified by sex, we found a statistically significant SRH–CRP association among men both in NP (β=0.27, 95% CI −0.03 to 0.57)

**Table 1** CRP values across characteristics of the study population

| | NP (n=646) | | CHARLS (n=8555) | |
|---|---|---|---|---|
| | Median (IQR)* | P value† | Median (IQR)* | P value† |
| Age | | <0.001 | | <0.001 |
| 45–60 | 0.6 (0.3 to 1.2) | | 0.9 (0.5 to 1.7) | |
| ≥60 | 0.8 (0.4 to 1.8) | | 1.1 (0.6 to 2.1) | |
| Sex | | 0.011 | | 0.003 |
| Men | 0.6 (0.3 to 1.3) | | 1.0 (0.5 to 2.0) | |
| Women | 0.8 (0.4 to 1.7) | | 0.9 (0.5 to 1.8) | |
| Education | | 0.004 | | 0.316 |
| Illiterate | 0.9 (0.4 to 1.8) | | 1.0 (0.5 to 2.0) | |
| Literate | 0.6 (0.3 to 1.3) | | 0.9 (0.5 to 1.9) | |
| Marital status | | 0.495 | | <0.001 |
| Married | 0.7 (0.3 to 1.5) | | 0.9 (0.5 to 1.9) | |
| Non-married | 0.7 (0.4 to 1.7) | | 1.1 (0.6 to 2.4) | |
| Smoking | | 0.467 | | 0.041 |
| Current smokers | 0.6 (0.3 to 1.4) | | 1.0 (0.5 to 2.0) | |
| Non-current smokers | 0.7 (0.4 to 1.6) | | 0.9 (0.5 to 1.9) | |
| Alcohol consumption | | 0.001 | | 0.635 |
| Regular drinkers | 0.5 (0.3 to 1.1) | | 0.9 (0.5 to 1.9) | |
| Non-regular drinkers | 0.8 (0.4 to 1.6) | | 1.0 (0.5 to 1.9) | |
| BMI | | <0.001 | | <0.001 |
| Underweight (<18.5) | 0.5 (0.2 to 1.4) | | 0.8 (0.5 to 1.9) | |
| Normal weight (18.5–24.99) | 0.6 (0.3 to 1.1) | | 0.8 (0.5 to 1.7) | |
| Overweight (25–29.99) | 1.2 (0.6 to 2.3) | | 1.2 (0.7 to 2.3) | |
| Obese (≥30) | 1.6 (1.0 to 4.4) | | 1.9 (0.9 to 3.3) | |
| Self-rated health | | 0.071 | | <0.001 |
| Good | 0.6 (0.3 to 1.7) | | 0.9 (0.5 to 1.8) | |
| Average | 0.7 (0.3 to 1.5) | | 0.9 (0.5 to 1.8) | |
| Poor | 0.8 (0.4 to 1.5) | | 1.0 (0.6 to 2.1) | |
| Very poor | 1.0 (0.5 to 2.3) | | 1.1 (0.6 to 2.3) | |
| Health status‡ | | 0.002 | | <0.001 |
| Healthy | 0.5 (0.3 to 1.3) | | 0.8 (0.5 to 1.7) | |
| Unhealthy | 0.8 (0.4 to 1.6) | | 1.0 (0.5 to 2.0) | |

Missing values: NP: 1 missing in health status. CHARLS: 2 missing in age, 7 missing in sex, 4 missing in education,1 missing in smoking, 3 missing in alcohol consumption, 1191 missing in BMI, 65 missing in health status.

*Median (Interquartile range, IQR).

†Analysis of variance (ANOVA) was applied to compare the mean of log-transformed values of CRP.

‡Health status: Unhealthy: Self-reported moderate to severe symptoms in the last month or used antihypertensive or antidiabetic medications (NP); Had been diagnosed by a doctor with any disease or often suffered from any pain currently (CHARLS). Healthy: no such report.

BMI, body mass index; CHARLS, China Health and Retirement Longitudinal Study; CRP, C-reactive protein; NP, Nanping project.

and CHARLS (β=0.12, 95% CI 0.05 to 0.19), but not in women. In a stratified analysis by education, the association between SRH and CRP was seen in literate people both in NP (β=0.26, 95% CI 0.02 to 0.51) and CHARLS (β=0.11, 95% CI 0.05 to 0.16), but not in illiterate people.

In the pooled population, the SRH–CRP association was repeated in the middle-aged (β=0.08, 95% CI 0.02 to 0.14), older people (β=0.08, 95% CI 0.02 to 0.15), men (β=0.13, 95% CI 0.06 to 0.20) and literate people (β=0.12, 95% CI 0.06 to 0.18) (figure 2).

**Additional analyses**

Identical trends with respect to the modifying effect of age and sex on the association between SRH and CRP were observed among literate people, but not among illiterate people (online supplementary table S3).

**Table 2** Association between self-rated health and C-reactive protein

| | N | Model 1* | | Model 2† | |
|---|---|---|---|---|---|
| | | β (95% CI) | P value | β (95% CI) | P value |
| NP | | | | | |
| Good health | 188 | Ref. | | Ref. | |
| Average | 270 | −0.03 (−0.22 to 0.17) | 0.792 | −0.05 (−0.24 to 0.14) | 0.589 |
| Poor | 165 | 0.12 (−0.10 to 0.34) | 0.292 | 0.10 (−0.11 to 0.32) | 0.349 |
| Very poor | 23 | 0.39 (−0.07 to 0.85) | 0.093 | 0.29 (−0.15 to 0.73) | 0.202 |
| Good/Poor‡ | 458/188 | 0.17 (-0.01 to 0.35) | 0.067 | 0.16 (−0.02 to 0.34) | 0.077 |
| CHARLS | | | | | |
| Good health | 1794 | Ref. | | Ref. | |
| Average | 4157 | 0.01 (-0.04 to 0.06) | 0.613 | 0 (−0.05 to 0.06) | 0.911 |
| Poor | 2157 | 0.10 (0.04 to 0.15) | 0.001 | 0.06 (0 to 0.12) | 0.055 |
| Very poor | 447 | 0.16 (0.06 to 0.25) | 0.001 | 0.11 (0.01 to 0.22) | 0.036 |
| Good/Poor | 5951/2604 | 0.10 (0.05 to 0.14) | <0.001 | 0.07 (0.02 to 0.11) | 0.004 |
| NP+CHARLS | | | | | |
| Good health | 1982 | Ref. | | Ref. | |
| Average | 4427 | 0.02 (−0.03 to 0.07) | 0.379 | 0.01 (−0.04 to 0.06) | 0.643 |
| Poor | 2322 | 0.11 (0.05 to 0.16) | <0.001 | 0.08 (0.02 to 0.14) | 0.013 |
| Very poor | 470 | 0.18 (0.09 to 0.28) | <0.001 | 0.14 (0.04 to 0.24) | 0.007 |
| Good/Poor | 6409/2792 | 0.11 (0.06 to 0.15) | <0.001 | 0.08 (0.03 to 0.12) | 0.001 |

*Adjusted for age, sex, education.
†Adjusted for age, sex, education, marital status, smoking, alcohol consumption, BMI, health status.
‡Good=Good+Average, Poor=Poor+Very poor.
CHARLS, China Health and Retirement Longitudinal Study; NP, Nanping project.

## DISCUSSION

In this study, based on 9201 residents in rural area of China, we found that poor SRH was associated with an elevated level of CRP in middle-aged and older people, especially among men and literate people.

Our finding of the association between poorer SRH and higher CRP level was in line with results from previous studies that included participants at similar age as our study participants.[11 14] Yet, those studies mainly included people living in industrialised countries with higher education, while our participants resided in less developed country with features of low literacy.

Possible pathways linking poor SRH and an elevated level of CRP could be related to psychological stress and health behaviours. Poor SRH may reflect a poor physical (e.g., inaccessibility to health service) and social (e.g., limited social network) environment, which can limit one's coping ability and induce psychological stress. It is known that stress can activate the sympathetic nervous system and the hypothalamic-pituitary-adrenal axis, contributing to the production of stress hormones, which in turn increase the secretion of CRP.[29 30] In addition, people with poor SRH were less likely to have an active lifestyle.[31] Having an inactive lifestyle has been suggested to potentially weaken the immune system and facilitate the inflammation processes through the release of pro-inflammatory adipokines.[32]

It is notable that poor SRH was associated with an elevated CRP level in literate participants, but not in illiterate participants, which was consistent with one previous study.[21] Similar findings were also shown in studies focusing on SRH and mortality.[18 19] One of the possible explanations may be that illiterate people are often lack of health-related knowledge and access to healthcare,[17] and thus may misinterpret the feeling that they have in their bodies.[33] It has been shown that poor SRH in the less educated people mainly represents less serious diseases.[34] In our study, we also found that illiterate people were more likely to rate their health as poor and to report illness or pain both in NP and CHARLS. Moreover, illiterate people may have to withstand more pressure as they have less social and financial resources. Thus, other factors may contribute to the reported poor SRH, rather than actual health condition.

We found that SRH–CRP associations were only observed in men, but not in women, which may be due to the potential sex differences in reporting SRH. Previous studies have shown that the poor SRH in women can reflect both serious and non-serious diseases, whereas it tends to reflect serious diseases in men.[35] Broad dimensions of health perceptions may lead to less accurate SRH in women. In addition, the proportion of illiterate people among women is much higher than that among men in

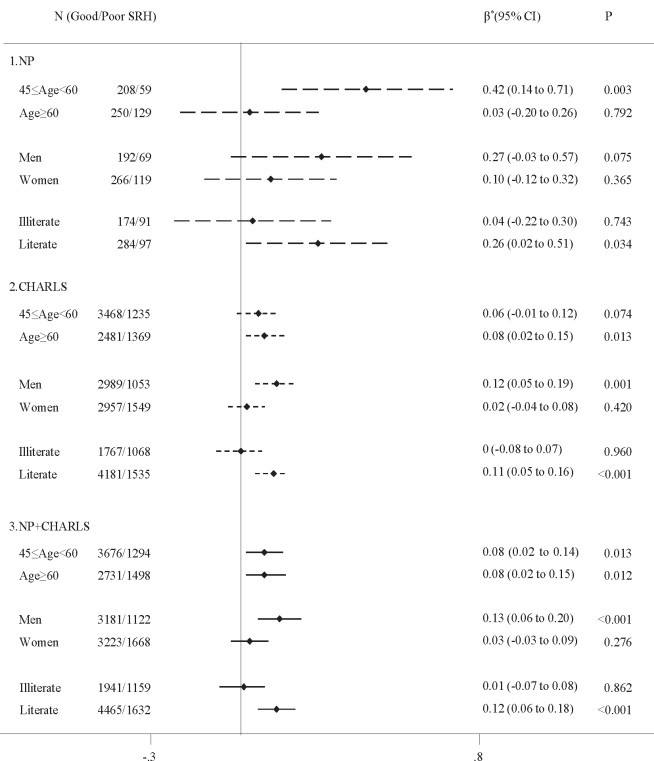

| | N (Good/Poor SRH) | | β*(95% CI) | P |
|---|---|---|---|---|
| **1.NP** | | | | |
| 45≤Age<60 | 208/59 | | 0.42 (0.14 to 0.71) | 0.003 |
| Age≥60 | 250/129 | | 0.03 (-0.20 to 0.26) | 0.792 |
| | | | | |
| Men | 192/69 | | 0.27 (-0.03 to 0.57) | 0.075 |
| Women | 266/119 | | 0.10 (-0.12 to 0.32) | 0.365 |
| | | | | |
| Illiterate | 174/91 | | 0.04 (-0.22 to 0.30) | 0.743 |
| Literate | 284/97 | | 0.26 (0.02 to 0.51) | 0.034 |
| **2.CHARLS** | | | | |
| 45≤Age<60 | 3468/1235 | | 0.06 (-0.01 to 0.12) | 0.074 |
| Age≥60 | 2481/1369 | | 0.08 (0.02 to 0.15) | 0.013 |
| | | | | |
| Men | 2989/1053 | | 0.12 (0.05 to 0.19) | 0.001 |
| Women | 2957/1549 | | 0.02 (-0.04 to 0.08) | 0.420 |
| | | | | |
| Illiterate | 1767/1068 | | 0 (-0.08 to 0.07) | 0.960 |
| Literate | 4181/1535 | | 0.11 (0.05 to 0.16) | <0.001 |
| **3.NP+CHARLS** | | | | |
| 45≤Age<60 | 3676/1294 | | 0.08 (0.02 to 0.14) | 0.013 |
| Age≥60 | 2731/1498 | | 0.08 (0.02 to 0.15) | 0.012 |
| | | | | |
| Men | 3181/1122 | | 0.13 (0.06 to 0.20) | <0.001 |
| Women | 3223/1668 | | 0.03 (-0.03 to 0.09) | 0.276 |
| | | | | |
| Illiterate | 1941/1159 | | 0.01 (-0.07 to 0.08) | 0.862 |
| Literate | 4465/1632 | | 0.12 (0.06 to 0.18) | <0.001 |

**Figure 2** β-coefficient and 95% CI of CRP in relation to poor self-rated health from linear regression models stratified by age, sex and education in NP, CHARLS, and the pooled populations of the two data sets. SRH is dichotomised as poor to very poor versus good to average. When stratified by age, models are adjusted for sex, education, marital status, smoking, alcohol consumption, BMI, health status; when stratified by sex, models are adjusted for age, education, marital status, smoking, alcohol consumption, BMI, health status; when stratified by education, models are adjusted for age, sex, marital status, smoking, alcohol consumption, BMI, health status. *The average CRP changes in response to one-unit shift in SRH. BMI, body mass index; CHARLS, China Health and Retirement Longitudinal Study; NP, Nanping project; SRH, self-rated health.

both data sets. This may explain the inconsistent findings between our study (6% participants with more than 9 years of schooling) and the Iwate-KENCO study from Japan, in which the corresponding figure was 46%.[14]

Findings from two data sets were not completely consistent. The association between poor SRH and elevated CRP values among older people (aged ≥60 years) was observed in CHARLS, but not in NP. In both populations, poor SRH was only associated with higher CRP in men, not in women. One of the explanations for these findings may be related to educational levels in the two study populations. Indeed, the proportion of illiterate people was relatively higher in older adults in NP (76.2%) than in CHARLS (58.3%), and there was a higher proportion of illiterate people in women in both populations. Second, we observed similar age and sex differences in the associations between SRH and CRP among the literate: poor SRH was associated with elevated CRP values, especially in men, which was the same as the main results. This

suggests that education might play a role in the SRH–CRP association.

The strengths of this study include the objective measure of CRP, the use of two different study populations to increase the confidence of our findings, and the high generalisability of our results to rural population of China given the use of national representative sample, CHARLS.

There are several limitations that should be considered. First, the cross-sectional study design prevented us from making causal inferences. Second, CRP was evaluated using different methods in NP and CHARLS. Nevertheless, the association between SRH and CRP did not differ between the two cohorts. Third, the self-reported SRH and some of the covariates may introduce reporting bias. Fourth, selection bias may arise due to the use of convenience sampling in NP. However, the results from NP were similar to those from CHARLS, which is a national representative sample. Finally, residual confounding or hidden bias cannot be ruled out due to lack of information on some potential confounders, such as clinical cardiovascular risk factors (e.g., High-density lipoprotein cholesterol (HDL-C); Hemoglobin A1c (HbA1c)), acute inflammatory conditions and medication use.

This study provides evidence that SRH, a simple measurement, may be used as an indicator of bad physical health among middle-aged and older literate people, but not among the illiterate people, in rural area. In China, the implementation of health surveillance is more challenging in rural than in urban areas because of the discrepant ageing processes,[36] knowledge gaps[22] and income inequality between these two areas. Elevated CRP has been associated with various physical[1–4] and psychological health outcomes.[37 38] Thus, our results support the consideration of using an efficient and cost-effective way, such as SRH, to monitor the health status in rural population where medical resources are limited. Future studies are needed to confirm our results and extend these findings to larger and more diverse populations. Moreover, identification of simple health indicators for illiterate people is warranted.

**Author affiliations**
[1]Department of Community Nursing, School of Nursing, Yangzhou University, Yangzhou, China
[2]Department of International Health and Medical Anthropology, Institute of Tropical Medicine (NEKKEN), Nagasaki University, Nagasaki, Japan
[3]Leading Program, Graduate School of Biomedical Sciences, Nagasaki University, Nagasaki, Japan
[4]Aging Research Center, Department of Neurobiology, Care Sciences and Society (NVS), Karolinska Institutet and Stockholm University, Stockholm, Sweden
[5]Department of Public Health, Nagasaki Prefectural Institute of Environment and Public Health, Nagasaki, Japan
[6]Stress Research Institute, Faculty of Social Sciences, Stockholm University, Stockholm, Sweden

**Acknowledgements** We would like to express our sincere gratitude to the participants and local staff in Nanping project (NP). We are grateful for those supporters: Harvard University (Aki Yazawa); National Center for Global Health and Medicine, Japan (Yosuke Inoue); Nagasaki Prefectural Institute of Environment and Public Health (Guoxi Cai); Fujian Medical University (Fei He, Jie Chen); Fujian

Provincial Center for Disease Control and Prevention (Meng Huang) during the data collection in NP. Data from China Health and Retirement Longitudinal Study (CHARLS) were collected by the National School of Development at Peking University, China. We appreciate the University of Copenhagen (Tianwei Xu); Fujian Provincial Center for Disease Control and Prevention (Xiuquan Lin); Nagasaki University (Sabin Nundu) for providing valuable comments in analysis and interpretation of data.

**Contributors** HXW, RPT and KYP conceptualised the study. RPT analysed the data and drafted the manuscript. HXW, KYP, GXC and TY contributed to critical revisions of the manuscript. RPT and HXW are responsible for ensuring the integrity and accuracy of the study. All authors have read and approved the final manuscript.

**Funding** This study is financed by the Program for Nurturing Global Leaders in Tropical and Emerging Communicable Diseases, Graduate School of Biomedical Sciences, Nagasaki University, Japan (RPT); the Swedish Research Council (Grant no: 2018-02998) and the Swedish Research Council for Health, Working Life and Welfare (Forte) (2019-01120) (HXW); the Ministry of Education of Taiwan, the Swedish National Graduate School on Ageing and Health (SWEAH), and Gamla Tjänarinnor Foundation (2016-00358) (KYP). NP was financially supported by the JSPS KAKENHI from the Japan Society for the Promotion of Science (13J06172).

**Competing interests** None declared.

**Patient consent for publication** Not required.

**Ethics approval** The Ethics Committee for Medical Research at the University of Tokyo (No. 10515-(1)) and the Ethics Committee of the Institute of Tropical Medicine at Nagasaki University (No.120910100-5) approved the study protocol of NP. The Medical Ethics Committee of Peking University approved the research protocol of CHARLS.

**Provenance and peer review** Not commissioned; externally peer reviewed.

**Data availability statement** All of the CHARLS data will be accessible to researchers around the world at the CHARLS project website (http://charls.pku.edu.cn/en). No additional data available.

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
