## [Reviewer comments · BMJ Open]

ARTICLE DETAILS

TITLE (PROVISIONAL)	The role of education in the association between self-rated health and levels of C-reactive protein: a cross-sectional study in rural areas of China
AUTHORS	Tu, Raoping; Pan, Kuan-Yu; Cai, Guoxi; Yamamoto, Taro; Wang, Hui-Xin

VERSION 1 – REVIEW

REVIEWER	Woojae Myung Department of Neuropsychiatry, Seoul National University Bundang Hospital.
REVIEW RETURNED	12-Nov-2018

GENERAL COMMENTS	1. Please show the full data of missing CRP values (line 40)2. The correlations they found between CRP and self-rated health could all be accounted for by individuals who had infections at the time of sampling. I think the author could show the result excluding subjects who had symptoms of acute infection, at least as sensitivity analysis.3. I think that the authors can expand their discussion with some previous studies, such as larger cohort study that investigated the association between hs-CRP and depressive symptom or psychological distress symptom, or genetic studies for subjective well-being.4. Is there any reason to combine "good " and "average" as good group, "poor" and "very poor" as poor group. I feel that it is an arbitrary classification.5. Why they did not show four group analysis (good/average/poor/very poor) in pooled samples (NP+CHARLS) in table 2?
--

REVIEWER	Zuyun Liu Yale University, USA
REVIEW RETURNED	22-Nov-2018

GENERAL COMMENTS	In this study, the authors aimed to explore the association between SRH and CRP among middle-aged and older Chinese adults in rural areas, in full sample and by age, sex, and education. The results suggested that poor SRH is a predictor of elevated CRP levels in this study population, especially in men and literate subpopulation. The MS is well-written. The analyses
--

	are well performed. I only have one major comment and some minor comments for the authors to consider. Major: It is confusing why the authors included two cohorts/datasets with different study design in this study. NP is a voluntary participants study, does it mean no rigorous sampling method? CHARLS is a nationally representative longitudinal study. How can the two different studies without any potential connection be included in one project? Minor:  1. NP is rural-based or it covers both rural and urban? 2. Although the authors have done MI for missing data issue for CRP, it is good to present the results instead simply saying “data not shown”, because the prop. of missing data on CRP is about 22% in CHARLS. Also the difference between persons with CRP and those without CRP should be clearly presented, e.g., as Tables. 3. In methods section, “health status” is slightly confusing. Are there any chronic conditions such diabetes, hypertension that can be accounted for here? Or the author should distinguish it from SRH. 4. Table 1, do not think “F” column is needed, not common in Medicine journals. 5. Why SES is not included in the main analysis, rather than in the additional analysis? Any special concern? 6. The discussion is not well organized, should be improved largely. Hard to follow. It requires a lot of work. 7. some abbreviations should be checked, like OR, no need to repeat again and again. 8, Page 16, “Possibly this sex-differential finding was bound to the differences in reporting SRH by sex.” What do the author want to say here? 9. Table S1, the 3rd column is confusing, it is not SRH, it is OR? 10. Table S2, is not a stratified by age, sex, but only by education.
--	--

VERSION 1 – AUTHOR RESPONSE

Reviewer #1: Woojae Myung

1. Please show the full data of missing CRP values (line 40)

We are grateful to reviewer for pointing out this weakness. As suggested, we have added two tables (Table S1 and Table S2) to show the full data of missing in CRP values in the NP and CHARLS separately.

2. The correlations they found between CRP and self-rated health could all be accounted for by individuals who had infections at the time of sampling. I think the author could show the result excluding subjects who had symptoms of acute infection, at least as sensitivity analysis.

We thank the reviewer for this helpful comment! We agree that the result excluding people who had symptoms of acute infection would be more reliable. In the current study, we have already excluded those whose CRP>10 mg/L (NP: n=25; CHARLS: n=429) due to potential acute inflammatory conditions. (Figure1)

3. I think that the authors can expand their discussion with some previous studies, such as larger cohort study that investigated the association between hs-CRP and depressive symptom or psychological distress symptom, or genetic studies for subjective well-being.

We sincerely thank the reviewer's precious advice! As suggested, we have now underlined the predictive role of CRP on psychological health by using two studies as references (p.16, line 8-10), one is a Danish cohort study consisting 73131 individuals with 20 years follow up, and another one is a systematic review of eight longitudinal studies. Now it reads: "By contrary, elevated CRP has been linked to depressive symptom or psychological distress symptom, which may also result in poorly rated health status in individuals". Meanwhile, we interpret the practical significance of SRH-CRP association in conclusion part by using these two references as well (p.17, line 25; p.18, line 1-5). Now it reads: "In China, the implementation of health surveillance is more challenging in rural than in urban areas because of the discrepancy in the aging processes, knowledge gaps and income inequality between these two areas. Elevated CRP has been associated with various physical and psychological health outcomes. Thus, our results support the consideration of using an efficient and cost-effective way, such as SRH, to monitor the health status in rural population where medical resources are limited".

4. Is there any reason to combine "good " and "average" as good group, "poor" and "very poor" as poor group. I feel that it is an arbitrary classification.

We thank the reviewer's comment! We observed the same pattern in both two datasets that poor and very poor SRH demonstrated similar effect of estimating CRP and so as good and average SRH. To simplify interpretation of the results, we decided to combine "good" and "average" as good group and "poor" and "very poor" as poor group. We have stated this reason on p.13, line 6-12. Now it reads: "Despite insignificance, the estimated effect of SRH started to change direction from average SRH ($\beta = -0.05$) to poor SRH ($\beta = 0.10$). In CHARLS, poor and very poor SRH were both associated with higher CRP ($\beta=0.06$, 95%CI 0 to 0.12; $\beta=0.11$, 95%CI 0.01 to 0.22). Considering the same pattern in both two datasets that poor and very poor SRH have similar effect on CRP and so as good and average SRH, and that there are limited number of participants with very poor SRH in NP, we combined 'good' and 'average' as good SRH, 'poor' and 'very poor' as poor SRH".

5. Why they did not show four group analysis (good/average/poor/very poor) in pooled samples (NP+CHARLS) in table 2?

Thank you for the concern! We agree with the reviewer that four group analysis (good/average/poor/very poor) in pooled samples (NP+CHARLS) can provide more information in understanding the SRH-CRP association. Therefore, we have added the result in Table2.

Reviewer #2: Zuyun Liu

Major:

It is confusing why the authors included two cohorts/datasets with different study design in this study. NP is a voluntary participants study, does it mean no rigorous sampling method? CHARLS is a nationally representative longitudinal study. How can the two different studies without any potential connection be included in one project?

We sincerely appreciate the reviewer for the valuable comments. Our response to the questions is as follows:

We agree with the reviewer's that although in NP the included seven communities were representative all counties in Nanping city in terms of economic development and population size, a convenience voluntary sample is used in the project, which indeed is a limitation of this study (p.4, line14-15).

However, this study is a part of a PhD student's project that required the student to collect and use data (i.e., NP sample) in the study. Due to the limited representability in the NP, which may lead to low validity, we decided to confirm the results observed from the NP in a nationally representative longitudinal study (CHARLS), therefore we included two datasets in this study. Cross verification from two or more sources can promote data validation by using two or more different methods to measure the same phenomenon.¹ The application of these two samples may help draw more accurate estimation and increase our confidence in making the conclusion. Nevertheless, we observed similar results in both datasets, and the level of heterogeneity is very low between two datasets ($I^2 < 0.001\%$).

Minor:

1. NP is rural-based or it covers both rural and urban?

We thank the reviewer for pointing this out. NP is a rural-based study, which have been mentioned on p.3, line 9 and p.6, line 8.

2. Although the authors have done MI for missing data issue for CRP, it is good to present the results instead simply saying "data not shown", because the prop. of missing data on CRP is about 22% in CHARLS. Also the difference between persons with CRP and those without CRP should be clearly presented, e.g., as Tables.

We appreciate this suggestion! We have now added Table S4 and Table S5 to present the results from multiple imputation for missing CRP values, Table S1 and Table S2 to show the differences between people with and without information on CRP.

3. In methods section, "health status" is slightly confusing. Are there any chronic conditions such diabetes, hypertension that can be accounted for here? Or the author should distinguish it from SRH.

Thank you so much for catching this point! We agree that "health status" should be distinguished from SRH, therefore, we defined the "health status" on table notes: "Unhealthy: Self-reported moderate to severe symptoms in the last month or used antihypertensive or antidiabetic medications (NP); Had been diagnosed by a doctor with any disease or often suffered from any pain currently (CHARLS). Healthy: no such report". (p.12, Table1, b)

4. Table 1, do not think "F" column is needed, not common in Medicine journals.

Thank you for the comments! We have now deleted "F" column as suggested. (pp.11-12, Table1)

5. Why SES is not included in the main analysis, rather than in the additional analysis? Any special concern?

We acknowledge reviewer's concern. Our reasons for excluding SES in the main analysis are as follows:

The measurement of SES in two datasets are different. In NP, SES is a self-rated household income, which depends on the comparison of oneself to others, i.e. "How is your household income compared to other people in the village? (0 point for the poorest, 10 points for the richest people, how many

points do you have?)". In CHARLS, SES is a self-rated household living standards, i.e. "Overall, how would you rate your own standard of living? Response options were "very high, relatively high, average, relatively poor, and poor" (p.9, line 21-24).

6. The discussion is not well organized, should be improved largely. Hard to follow. It requires a lot of work.

We are grateful to reviewer for pointing out this weakness. We have now addressed the reviewer's concern by revising the discussion extensively (pp.15-17). Now it reads as follows:

"In this study, based on 9201 residents in rural area, we found that poor SRH is associated with an elevated level of CRP in middle-aged and older people, especially among the literate and men.

Our finding of the association between poorer SRH and higher CRP level is in line with results from previous studies that included participants in similar age as our study participants. Yet, those studies mainly looked at people living in industrialized countries with higher education while our participants resided in less developed country with features of low literacy.

Possible pathways linking poor SRH and elevated level of CRP could be due to psychological stress and health behavior. Poor SRH may reflect a poor physical (e.g., inaccessibility to health service) and social environment (e.g., limited social network), which can limit ones coping ability and induce psychological stress. It is known that stress can activate the sympathetic nervous system and the hypothalamic-pituitary-adrenal axis, contributing to the production of stress hormones, which in turn increase the secretion of CRP. In addition, people with poor SRH were less likely to have an active lifestyle. Having an inactive lifestyle has been suggested to potentially weaken the immune system and facilitate the inflammation processes through the release of pro-inflammatory adipokines. Furthermore, poor SRH may also reflects poor medication adherence, such as low aspirin adherence, which has been associated with elevated levels of CRP in the first 3 months after acute coronary syndrome. By contrary, elevated CRP has been linked to depressive symptom or psychological distress symptom, which may also result in poorly rated health status in individuals.

It is notable that poor SRH was associated with an elevated CRP level in literate participants, but not in the illiterate participants, which was consistent with one previous study. Indeed, similar results were also shown in studies focusing on SRH and mortality. One of the possible explanations may be that illiterate people are often lack of health-related knowledge and access to health care, and thus may misinterpret the feeling that they have in their bodies. It has been shown that poor SRH in the less educated people mainly represents less serious diseases. In our study, we also found that illiterate people were more likely to rate their health as poor and to report illness or pain both in NP and CHARLS (Supplementary File: Table S6). Moreover, illiterate people may have to withstand more pressure as they have less social and financial resources. Thus, other factors may contribute to the reported poor SRH, rather than actual health condition.

We found that SRH-CRP associations were only observed in men, but not in women, which may be due to the potential sex differences in reporting SRH. Previous studies have shown that the poor SRH in women can reflect both serious and non-serious diseases, whereas it tends to reflect serious diseases in men. Broad dimensions of health perceptions may lead to less accurate SRH in women. In addition, the proportion of illiterate people among women is much higher than that among men in both datasets, this may explain the different findings between our study and the Iwate-KENCO study from Japan.

The discrepant findings between two datasets are worthy of discussion. First, the association between poor SRH and elevated CRP values among older people (aged \geq 60 years) was observed in CHARLS,

but not in NP. And in both populations, poor SRH was only associated with higher CRP in men, not in women. These findings may also be explained by educational level in each subgroup. That is, the proportion of illiterate people was relatively higher in older adults in NP (76.2%) than in CHARLS (58.3%) as shown in Table S6 (Supplementary File), and there was a higher proportion of illiterate people in women in both populations. Second, after excluding the illiterate people, we observed similar age and sex differences in the associations between SRH and CRP among the literate, i.e. poor SRH is associated with elevated CRP values among literate people, especially in men, which was the same as the main results. This suggests that education might play a role in the SRH-CRP association. Third, similar results were observed in urban areas of CHARLS, and further adjusting for socioeconomic status (i.e. self-rated household income in NP, self-rated household living standards in CHARLS) did not change the SRH-CRP association (data not shown), suggesting socioeconomic status might not influence the SRH-CRP association.”

7. some abbreviations should be checked, like OR, no need to repeat again and again.

Thank you for the observation. We carefully checked the manuscript again. As suggested by reviewer, we delete the extra abbreviation notes for OR. (p.14, line 22)

8, Page 16, “Possibly this sex-differential finding was bound to the differences in reporting SRH by sex.” What do the author want to say here?

Thank you for the careful reading! We try our best to improve this sentence and make it clearly by rewriting it (p.16, line 23-24). Now it reads: “We found that SRH-CRP associations were only observed in men, but not in women, which may be due to the potential sex differences in reporting SRH.”

9. Table S1, the 3rd column is confusing, it is not SRH, it is OR?

Thanks for your correction. In our resubmitted manuscript, the third column header has been revised (Table S3). Now it reads: “OR (95% CI)^a”. In addition, for making the title of Table S3 more specific, we revise the title as well. Now it reads: “Table S3 Odds ratio and 95% confidence interval (95% CI) between poor self-rated health and levels of C-reactive protein: stratified by age, sex and education (pooled population, logistic)”.

10. Table S2, is not a stratified by age, sex, but only by education.

We sincerely thank the reviewer for careful reading. The title of Table S2 has been corrected (Table S6). Now it reads: “Table S6 Characteristics of the study samples: stratified by datasets and education”.

Other modifications:

1. We made some modifications to Figure 2.

Removed the word 'population'.

Changed N(Good/Poor) to N (Good/Poor SRH)

Changed the legend to: β -coefficient and 95% confidence interval (CI) of CRP in relation to poor self-rated health from linear regression models stratified by age, sex and education in NP, CHARLS, and the pooled populations of the two datasets. SRH is dichotomized as poor to very poor versus good to average. Models are simultaneously adjusted for age, sex, education, marital status, smoking, alcohol consumption, BMI, health status.

2. We updated the Hui-Xin Wang’s funding source (p.19, line 3-4). Now it reads: “the Swedish Research Council (grant number: 2018-02998, Hui-Xin Wang)”.

3. We formatted the references and correct the order accordingly.
4. The order of supplementary tables is now corrected.
5. The word count is updated.

References

1. Choe, E. K., Lee, N. B., Lee, B., Pratt, W., & Kientz, J. A. (2014, April). Understanding quantified-selfers' practices in collecting and exploring personal data. In Proceedings of the 32nd annual ACM conference on Human factors in computing systems (pp. 1143-1152). ACM.

VERSION 2 – REVIEW

REVIEWER	Woojae Myung Department of Neuropsychiatry, Seoul National University Bundang Hospital, Seongnam, South Korea
REVIEW RETURNED	26-Feb-2019

GENERAL COMMENTS	Thank you for your response.
------------------------------

REVIEWER	Zuyun Liu Yale University
REVIEW RETURNED	09-Mar-2019

GENERAL COMMENTS	Thank the authors so much for addressing my concerns. One minor issue is that, the number of missing data on CPR in NP is not correct in Table S1 (because N=3 was mentioned in Fig 1). Relevant to this issue, I may not be able to review Table S4 and Table S5 as I do not have expertise in handling missing data. Independent variables may not be suitable for imputation using MI due to some unintended errors i remember. I may be wrong and would much appropriate if experts could help with. For other parts, the authors have done a nice job.
---

REVIEWER	Failde Inmaculada Universidad de Cadiz
REVIEW RETURNED	03-May-2019

GENERAL COMMENTS	The paper analyzes the role of educational level in the association between self-rated health and level of C-Reactive protein in 2 rural population in China. The study has been carried out in a large sample of subjects from 2 different rural populations. In spite of his interests, the study presents some important methodological limitation. - In the Introduction of the manuscript, the authors do not justify the potential role of education in the relationship between the 2 variables analyzed (CRP-SRH) nor define the hypothesis they intend to demonstrate. What would be a plausible explanation for the potential effect of the educational level on the relationship between CRP and SHH? They should include a paragraph about the effect of other possible variables that could affect the relationship that is addressed in the study.
--

	The objective of the study should be formulated more concretely including what is really intended. - The methodology presents important limitations that make the validity of the results questionable. Why do they analyze 2 different populations in which different methods are used for the determination of CRP? What sensitivity and specificity do each of these methods have? . The exclusion of some subjects is based on the CRP level > 10mg / L in a population and > 6.25mg / L in another, is not well justified without providing information on the possible acute pathological processes that could be cause of these level. How was alcohol consumption measured? What about the tobacco? . Has been the BMI self-reported?? It would be necessary to include in the analysis the covariates that have been described in other studies (as included in the discussion) that may affect the relationships between the variables analyzed. In the discussion, there is excessive speculation about potential relationships that have not been addressed in the study. The limitations of the study should be analyzed more in deep in the manuscript
--	---

REVIEWER	Ana Paula Santana Coelho Almeida Universidade Federal do Espírito Santo - Brasil
REVIEW RETURNED	29-Jul-2019

GENERAL COMMENTS	General comments: Very relevant study theme, however the text is confused by the excessive information in the results. Information can be suppressed as a logistic regression analysis, for example. Specific comments: In the summary, present the purpose of the study as presented in the introduction. The introduction is appropriate. Methodology: For additional analyzes, report only those presented in tables. In my opinion, it would be sufficient as an additional analysis only the data-imputed analyzes. Too much information makes the manuscript difficult to read. Results: Table 1 should also present the sample characterization through the frequency of the variables. This table should also specify which test was used in the analysis. As described in the methodology it is understood that it was ANOVA, but the table has values in median. Table 2- It was confusing the combination of self-rated health categories in the same table. The table should be self-explanatory and in this case, it was confusing for the reader who needs to use the text to understand. I suggest putting in another table or adding explanation in the table footer. In stratified analyzes, as in Figure 2, the models cannot be adjusted by the stratification variable. These analyzes should be corrected.
--

	In my opinion, Figure 2 is hard to understand. What are the values -0.3 and 0.8? I don't think it's necessary to redo the analyzes with logistic regression. There was a lot of information, confusing to the reader. In table S6 put identification in the column in relation to the values - n (%). In my opinion, table 6 is not within the scope of the study objectives. Discussion In Line 31 pg 17 the authors report that an analysis was performed excluding illiterate people, but this analysis is not presented in the manuscript. I think it is important to present a separate analysis of the associated factors of the association between SRH and CRP for literate and illiterate people, as there is a suspicion that the level of education changes the effect of the association between SRH and CRP.
--	---

VERSION 2 – AUTHOR RESPONSE

Reviewer #1: Woojae Myung

Thank you for your response.

Thank you so much for your reply!

Reviewer #2: Zuyun Liu

Thank the authors so much for addressing my concerns. One minor issue is that, the number of missing data on CPR in NP is not correct in Table S1 (because N=3 was mentioned in Fig 1). Relevant to this issue, I may not be able to review Table S4 and Table S5 as I do not have expertise in handling missing data.

Thank you for pointing this out. We carefully checked the manuscript again and found that one participant had missing values both in CRP and SRH in the NP study. When we compared the “Characteristics of study sample in NP without and with missing values in CRP” (Table S1), we excluded those who had missing values in SRH beforehand, while in Figure1, we removed those who had missing values in CRP at first instead. We have now addressed the reviewer’s concern by excluding those had missing values both in CRP and SRH in Figure1 (NP study and CHARLS). Now it reads: “No information both on CRP and SRH (n=1)” and “No information both on CRP and SRH (n=46)”. We modified the text as well (p.7, line 2-3; line 14-15), it reads: “After further excluding people with missing information on CRP (n=2), SRH (n=25), and on both CRP and SRH (n=1)” and

“We further excluded people with missing data on CRP (n=3810), SRH (n=28), and on both CRP and SRH (n=46)”.

Independent variables may not be suitable for imputation using MI due to some unintended errors I remember. I may be wrong and would much appreciate if experts could help with. For other parts, the authors have done a nice job.

We agree with the reviewer. In fact, we did not impute missing values in independent variables, for example, SRH, as the proportion of people with missing data was very low (<0.5%). In contrast, the proportion of participants with missing data in the dependent variable (i.e., CRP) was 22%. Thus, we decided to conduct MI to address this issue. Nevertheless, as reported in Tables S1 & S2, the missingness of CRP was not at random. In this situation, according to Hughes et al. (2019), no information is gained from imputing the outcome. We therefore have now removed all of the statements in connection with MI. An additional limitation has now been added, which reads “Convenience sampling in the Nanping project and the relatively large proportion of CHARLS participants with missing values in CRP may have introduced bias.” (p.4, line 13-14)

Reviewer #3: Failde Inmaculada

The paper analyzes the role of educational level in the association between self-rated health and level of C-Reactive protein in 2 rural population in China.

The study has been carried out in a large sample of subjects from 2 different rural populations. In spite of his interests, the study presents some important methodological limitation.

- In the Introduction of the manuscript, the authors do not justify the potential role of education in the relationship between the 2 variables analyzed (CRP-SRH) nor define the hypothesis they intend to demonstrate. What would be a plausible explanation for the potential effect of the educational level on the relationship between CRP and SHH?

We appreciate the reviewer’s comment. We have added justification and hypothesis to our manuscript (p.5, line 19-25 and p.6, line 1-4). Now it reads: “As SRH measures personal perception of health, it can be influenced by other factors beyond the real health status. For example, people with different educational levels may have different perceptions of health. This education-related difference in perception of health may further play a role in the association between SRH and health outcomes. Indeed, a stronger association between SRH and mortality among higher educated than lower educated individuals has been shown in two studies. Since CRP has been recognized as an important predictor of mortality, education seems to modify its relationship with SRH. It is noteworthy that studies concerning the association between SRH and CRP were mostly conducted in developed countries where the study populations were relatively well educated. To our knowledge, no study has focused on the difference in the association between SRH and CRP between illiterate and literate people.”

We agree with the reviewer that it is important to discuss the potential effect of the educational level on the association between SRH and CRP. We have expanded this in the discussion part (p.16, line 2-9). It reads: "One of the possible explanations may be that illiterate people are often lack of health-related knowledge and access to health care, and thus may misinterpret the feeling that they have in their bodies. It has been shown that poor SRH in the less educated people mainly represents less serious diseases. In our study, we also found that illiterate people were more likely to rate their health as poor and to report illness or pain both in NP and CHARLS. Moreover, illiterate people may have to withstand more pressure as they have less social and financial resources. Thus, other factors may contribute to the reported poor SRH, rather than actual health condition."

They should include a paragraph about the effect of other possible variables that could affect the relationship that is addressed in the study.

We thank the reviewer for this helpful comment! We agree that including other variables that may influence the association would be more comprehensive. Indeed, in addition to education, we previously have also included age and sex, according to the literature. We have now made these statements clearer in p.5, line 9-17. Now it reads: "These discrepancies may be due to differences in characteristics of the study populations (e.g., age and sex) and study design. For example, a Japanese study demonstrated an association between poor SRH and an elevated CRP value in women, but not in men (age range 40-69). In contrast, in an US sample of younger adults (mean age 28.42 ± 1.78), current SRH was not associated with CRP in women, whereas the association was shown in men. Among hospital-based studies, poor SRH was associated with higher CRP in female patients with coronary heart disease, but not in patients with breast cancer. In community-based studies, there has been a cross-sectional association between SRH and CRP, but no evidence indicating longitudinal association."

The objective of the study should be formulated more concretely including what is really intended.

We appreciate this suggestion! We have now formulated the objective more concretely (p.6, line 10-13), it reads: "In the current study, we use two databases from China to examine the association between SRH and CRP among middle-aged and older people in rural areas, and to explore whether the SRH-CRP association varies across age (45-60/ ≥ 60), sex (men/women), and educational levels (illiterate/literate)."

- The methodology presents important limitations that make the validity of the results questionable. Why do they analyze 2 different populations in which different methods are used for the determination of CRP? What sensitivity and specificity do each of these methods have? .

We thank the reviewer's comment. This study was involved in a PhD student's project that required the student (i.e., the first author) to collect and use the data (i.e., NP sample) in the study. As NP is a small and convenience sample, in order to overcome the limitation of the NP study design, to increase the study power and to strengthen the generalizability of the findings, we decided to include a large

and national representative sample from CHARLS in the current study. It has been suggested that cross verification from two or more sources can promote data validation by using two or more different methods to measure the same phenomenon.² The application of these two samples may help draw more accurate estimation and increase our confidence in making the conclusion. Indeed, in these two populations, CRP was not measured using the same method. (p.7, line 25, p.8, line 1-2; p.8, line 8-9). Although different measures of CRP were used in the two studies, we observed similar results in both cohorts. The estimate effects were in the same direction, and level of heterogeneity ($I^2 < 0.001\%$) is very small between two cohorts.

Nevertheless, CRP measurements in both populations (high sensitivity sandwich enzyme immunoassay in NP and immunoturbidimetric assay in CHARLS) have been widely used in lower limit of detection of CRP (about 2mg/l).³ Specifically, in the NP study, CRP measurement followed the standardized protocol which devised by Brindle et al.. Within and between assay CVs (Coefficient of Variation) were 3.4% and 9.9%, respectively. Assessments of assay limits of detection, linearity, recovery, imprecision, and concordance with an established method (Pearson correlation=0.988, n=20) demonstrated the validity.⁴ For the CHARLS, within and between assay CVs were <1.3% and <5.7%, and the detection limit was 0.1 to 20 mg/l.⁵ To ensure the accuracy of serum measurements, appropriate quality control and a Quality Assurance protocol were followed in National Health and Nutrition Examination Survey.⁶

The exclusion of some subjects is based on the CRP level > 10mg / L in a population and > 6.25mg / L in another, is not well justified without providing information on the possible acute pathological processes that could be cause of these level.

We chose to use different cut-offs for exclusion subjects who might have acute inflammation because the CRP levels were measured differently in the two databases. CRP levels were measured using dried blood spots in the NP, while levels of CRP were obtained from serum in the CHARLS. According to Brindle et al. 4, CRP in serum was on average 1.6 times higher than in dried blood spots (DBS). This means that those with CRP concentrations higher than 6.25 mg/L in DBS is comparable to 10 mg/L at serum level, which is a suggested cutoff point for potential acute inflammatory conditions.⁷ Therefore, we have chosen these two specific cut-offs to exclude participants with potential acute inflammatory conditions. We have now clarified the reasons of using two different cut-offs in p.6 line 25, p.7, line 1-2.

We agree with the reviewer that including the information on the potential acute inflammatory conditions will be very helpful. However, such information was not available in the NP or the CHARLS. We have now acknowledged this issue in the limitation (p.4, line 15-18), now it reads: "Residual confounding or hidden bias cannot be ruled out due to lack of information on some potential confounders, such as clinical cardiovascular risk factors (e.g, HDL-C, HbA1c), acute inflammatory conditions, medication use, etc."

How was alcohol consumption measured? What about the tobacco? . Has been the BMI self-reported??

Information concerning alcohol consumption, smoking and BMI has now been stated more clearly in p.8, line 25, p.9, line 1; p.8, line 24-25; p.8, line 14-15 and p.8, line 22-24, respectively. In both the NP and the CHARLS, alcohol consumption and smoking were self-reported using two questions: 1) frequency of drinking alcohol in one week and 2) whether they were current smoker. For BMI, height and weight were measured by interviewers using standard anthropometers and were used to calculate BMI.

It would be necessary to include in the analysis the covariates that have been described in other studies (as included in the discussion) that may affect the relationships between the variables analyzed.

We thank the reviewer's suggestion. We did try to look into other potential covariates that were included in other studies. Unfortunately, covariates such as total-c, hdl-c, hba1c were only available in the CHARLS but not in the NP. Information on medication use was limited to hypertension and diabetes in NP study. CHARLS had a considerable amount of missing data on current occupation (48%) and physical activity (61%). We have now acknowledged this issue in the limitation (p.4, line 15-18). Now it reads: "Residual confounding or hidden bias cannot be ruled out due to lack of information on some potential confounders, such as clinical cardiovascular risk factors (e.g, HDL-C, HbA1c), acute inflammatory conditions, medication use, etc."

In the discussion, there is excessive speculation about potential relationships that have not been addressed in the study.

We thank the reviewer's comment. We have now modified the third paragraph in the discussion (p.15, line 15-23). Now it reads: "Possible pathways linking poor SRH and an elevated level of CRP could be related to psychological stress and health behaviors. Poor SRH may reflect a poor physical (e.g., inaccessibility to health service) and social (e.g., limited social network) environment, which can limit one's coping ability and induce psychological stress. It is known that stress can activate the sympathetic nervous system and the hypothalamic-pituitary-adrenal axis, contributing to the production of stress hormones, which in turn increase the secretion of CRP. In addition, people with poor SRH were less likely to have an active lifestyle. Having an inactive lifestyle has been suggested to potentially weaken the immune system and facilitate the inflammation processes through the release of pro-inflammatory adipokines."

The limitations of the study should be analyzed more in deep in the manuscript

We appreciate the reviewer's comment. We have now listed the most relevant limitations (p.4 line 12-18) and complied the format required by the journal: strengths and limitations of this study contain up to five short bullet points, no longer than one sentence each that relate specifically to the methods.

Reviewer #4: Ana Paula Santana Coelho Almeida

General comments: Very relevant study theme, however the text is confused by the excessive information in the results. Information can be suppressed as a logistic regression analysis, for example.

We sincerely thank the reviewer for the valuable suggestion! To simplify the information of results and avoid confusion, we have now deleted "logistic regression analysis" related text and the old Table S3.

Specific comments:

In the summary, present the purpose of the study as presented in the introduction.

The introduction is appropriate.

Methodology:

For additional analyzes, report only those presented in tables. In my opinion, it would be sufficient as an additional analysis only the data-imputed analyzes. Too much information makes the manuscript difficult to read.

We thank the reviewer for the suggestion. We have now reduced information regarding additional analyses except the analysis of the SRH-CRP association for literate and illiterate people (p.9, line 22-24; p.15, line 1-3). In line with our response to reviewer #2, statements regarding imputation were also removed. It reads: "In order to compare our results with previous studies that including participant with formal education only, we performed additional linear regression analysis stratified by age and sex among illiterate and literate participants separately." "Identical trends with respect to the modifying effect of age and sex on the association between SRH and CRP were observed among literate people, but not among illiterate people (Supplementary File: Table S3)."

Results:

Table 1 should also present the sample characterization through the frequency of the variables. This table should also specify which test was used in the analysis.

As described in the methodology it is understood that it was ANOVA, but the table has values in median.

We thank the reviewer for the suggestion. We were aware that ANOVA cannot be used for comparing median values of CRP as it is not normally distributed (p.9, line 12), thus we log-transformed the CRP

values before applying ANOVA. We have now clarified it by adding a footnote to Table 1: b ANOVA was applied to compare the mean of log-transformed values of CRP (p.12, b).

Table 2- It was confusing the combination of self-rated health categories in the same table. The table should be self-explanatory and in this case, it was confusing for the reader who needs to use the text to understand. I suggest putting in another table or adding explanation in the table footer.

Thank you for the suggestion. We have now added a footnote under Table 2 as suggested, now it reads: "c Good= Good+Average, Poor=Poor+Very Poor" (p.14, line 1).

In stratified analyzes, as in Figure 2, the models cannot be adjusted by the stratification variable. These analyzes should be corrected.

We thank the reviewer for this helpful comment! We agree that the models cannot be adjusted by the stratification variable. Indeed, we excluded those stratification variables in stratified analyses. To clarify this issue, we have now added contents in figure notes (Figure 2). Now it reads: "Figure 2 β -coefficient and 95% confidence interval (CI) of CRP in relation to poor self-rated health from linear regression models stratified by age, sex and education in NP, CHARLS, and the pooled populations of the two datasets. SRH is dichotomized as poor to very poor versus good to average. When stratified by age, models are adjusted for sex, education, marital status, smoking, alcohol consumption, BMI, health status; when stratified by sex, models are adjusted for age, education, marital status, smoking, alcohol consumption, BMI, health status; when stratified by education, models are adjusted for age, sex, marital status, smoking, alcohol consumption, BMI, health status."

In my opinion, Figure 2 is hard to understand. What are the values -0.3 and 0.8?

-0.3 and 0.8 are the values of β , which indicates the degree of change in the mean values of CRP for one-unit increment in the SRH scores after adjusting other variables in the model. To clarify this, we have now added a footnote concerning the β values (Figure 2), it reads: "a The average CRP changes in response to one-unit shift in SRH."

I don't think it's necessary to redo the analyzes with logistic regression. There was a lot of information, confusing to the reader.

We appreciate for reviewer's valuable comments. We have now removed "logistic regression analysis" related text and old Table S3 simultaneously.

In table S6 put identification in the column in relation to the values - n (%). In my opinion, table 6 is not within the scope of the study objectives.

We thank the reviewer's suggestion. Now we have taken away Table S6.

Discussion

In Line 31 pg 17 the authors report that an analysis was performed excluding illiterate people, but this analysis is not presented in the manuscript. I think it is important to present a separate analysis of the associated factors of the association between SRH and CRP for literate and illiterate people, as there is a suspicion that the level of education changes the effect of the association between SRH and CRP.

We sincerely thank the reviewer's precious advice! As suggested, we have now added one table (Table S3) in our supplementary file.

Other modifications:

1. We updated Raoping Tu's affiliation (p.1, line 7).
2. We revised the setting and participants for making abstract clearer (p.3, line 6-9).
3. We changed "CRP a" to "Median (IQR) a" in Table 1, "Good" to "Good health" in Table 2.
4. The range of normal weight and overweight in BMI has been revised, it reads: "18.5-24.99", "25-29.99" (Table 1).
5. One statement has been made clearer (p.16, line 16-18). Now it reads: "This may explain the inconsistent findings between our study (6% participants with more than 9 years of schooling) and the Iwate-KENCO study from Japan, in which the corresponding figure was 46%."
6. We formatted the references and corrected the order accordingly.
7. The order of supplementary tables has been corrected.
8. The word count has been updated.

References

1. Hughes, R. A., Heron, J., Sterne, J. A., & Tilling, K. (2019). Accounting for missing data in statistical analyses: multiple imputation is not always the answer. *International journal of epidemiology*, 1, 11.
2. Choe EK, Lee NB, Lee B, et al. Understanding quantified-selfers' practices in collecting and exploring personal data. 2014:1143-52.
3. Aziz, N., Fahey, J. L., Detels, R., & Butch, A. W. (2003). Analytical performance of a highly sensitive C-reactive protein-based immunoassay and the effects of laboratory variables on levels of protein in blood. *Clin. Diagn. Lab. Immunol.*, 10(4), 652-657.
4. Brindle, E., Fujita, M., Shofer, J., & O'Connor, K. A. (2010). Serum, plasma, and dried blood spot high-sensitivity C-reactive protein enzyme immunoassay for population research. *Journal of immunological methods*, 362(1-2), 112-120.
5. Zhao Y, Crimmins E, Hu P, et al. CHARLS Blood Sample Users' Guide. National School of Development, Peking University, 2014.
6. Centers for Disease Control and Prevention. National Health and Nutrition Examination Survey Laboratory Procedures
https://www.cdc.gov/nchs/data/nhanes/nhanes_11_12/201112_laboratory_procedures_manual.pdf.
7. Carlson, C. S., Aldred, S. F., Lee, P. K., Tracy, R. P., Schwartz, S. M., Rieder, M., ... & Fornage, M. (2005). Polymorphisms within the C-reactive protein (CRP) promoter region are associated with plasma CRP levels. *The American Journal of Human Genetics*, 77(1), 64-77.

VERSION 3 – REVIEW

REVIEWER	Ana Paula Santana Coelho Almeida Universidade Federal do Espírito Santo. Brazil.
REVIEW RETURNED	20-Sep-2019
GENERAL COMMENTS	Congratulations on improvement in the paper.

VERSION 3 – AUTHOR RESPONSE

Reviewer #4: Ana Paula Santana Coelho Almeida

Congratulations on improvement in the paper.

Thank you so much for your reply!

Other modifications:

1. We updated Raoping Tu's affiliation (p.1, line 8-9).
2. The word count has been updated.